# In Silico Discovery of Anticancer Peptides from Sanghuang

**DOI:** 10.3390/ijms232213682

**Published:** 2022-11-08

**Authors:** Minghao Liu, Jiachen Lv, Liyuan Chen, Wannan Li, Weiwei Han

**Affiliations:** Key Laboratory for Molecular Enzymology and Engineering of Ministry of Education, School of Life Science, Jilin University, 2699 Qianjin Street, Changchun 130012, China

**Keywords:** Sanghuang, reverse docking, anti-cancer peptide prediction, quantum chemical calculation

## Abstract

Anticancer peptide (ACP) is a short peptide with less than 50 amino acids that has been discovered in a variety of foods. It has been demonstrated that traditional Chinese medicine or food can help treat cancer in some cases, which suggests that ACP may be one of the therapeutic ingredients. Studies on the anti-cancer properties of *Sanghuangporus sanghuang* have concentrated on polysaccharides, flavonoids, triterpenoids, etc. The function of peptides has not received much attention. The purpose of this study is to use computer mining techniques to search for potential anticancer peptides from 62 proteins of Sanghuang. We used mACPpred to perform sequence scans after theoretical trypsin hydrolysis and discovered nine fragments with an anticancer probability of over 0.60. The study used AlphaFold 2 to perform structural modeling of the first three ACPs discovered, which had blast results from the Cancer PPD database. Using reverse docking technology, we found the target proteins and interacting residues of two ACPs with an unknown mechanism. Reverse docking results predicted the binding modes of the ACPs and their target protein. In addition, we determined the active part of ACPs by quantum chemical calculation. Our study provides a framework for the future discovery of functional peptides from foods. The ACPs discovered have the potential to be used as drugs in oncology clinical treatment after further research.

## 1. Introduction

*Sanghuangporus sanghuang* is a precious medicinal fungus. Both the medicinal and edible use of Sanghuang have a long history in China [1]. Sanghuang plays an effective role in the treatment of cervical cancer manifested by abnormal vaginal bleeding [2]. Most of the existing Sanghuang studies on various diseases focus on polysaccharides, exopolysaccharides, and other bioactive substances [1,3]. For instance, a polysaccharide extracted from Sanghuang has proven antidiabetic activity [4]. The anti-tumor activity of Sanghuang was first reported in 1968 [5]. The anti-cancer function of Sanghuang has now been confirmed by many studies. Extract of Sanghuang induced cell death in human cervical cancer SiHa cells, both in vitro and in vivo [2]. As a traditional Chinese medicine and food, the anticancer activity of Sanghuang may be related to the anticancer peptides. However, the study of the anticancer mechanism of Sanghuang peptides has been neglected.

Anticancer peptide [6], also called ACP, is a newly emerging cancer therapy. ACPs exhibit cancer cell-selective toxicity [7]. Most anticancer peptides are derived from antimicrobial peptides. Similar to antimicrobial peptides, ACPs bind to receptors through electrostatic interactions [8]. The mechanisms of anticancer peptides include altering cell membrane permeability and interacting with cancer cell endogenous targets [9]. Anticancer peptides are traditionally discovered using experimental methods. However, biological experiments are time-consuming and inefficient. With the support of a large amount of protein sequence information, many computer prediction methods have emerged, as the times require [10]. 

The current anticancer peptide prediction models include two types of computer algorithms based on traditional machine learning and deep learning [11]. Among them, most of the traditional machine learning algorithms are based on support vector machines. AntiCP is the first computational predictive model based on sequence feature descriptors. AntiCP2.0 is designed to predict and design anti-cancer peptides with high accuracy. The main dataset consists of 861 experimentally validated anticancer peptides and 861 non-anticancer or validated antimicrobial peptides [12]. The online site also offers a protein scan feature that can identify regions in proteins. These regions facilitate the design of anticancer peptides, which can be further cleaved or edited. The SVM machine algorithm is adopted by mACPpred. Peptides were screened by mACPpred to identify ACPs [13]. The anticancer peptide data used in this paper were obtained from the CancerPPD database [14]. This online site is composed of 3,491 ACPs and 121 anticancer protein entries collected by the authors from existing articles and patents. The information includes the source of the anticancer peptide, the properties of the peptide, anticancer activity, N-terminal and C-terminal modifications, and conformation, etc. [14]. By using reverse docking [15], we want to identify the ACP target proteins, after which we will display their interactions using visualization tools.

These bioinformatics methods were used to predict anticancer peptides in Sanghuang. The workflow is shown in Figure 1. The anticancer potential of Sanghuang protein and trypsin-digested fragments was predicted by mACPpred [13]. Our study used AntiCP 2.0 to predict the charge of possible ACPs and to demonstrate their anticancer potential from a mechanism perspective [16]. Through the blast function of the CancerPPD database, a known ACP with high similarity to one of the sequences and its ligand was found [14]. The structure of this peptide was simulated and HDOCK was used to perform molecular docking [17]. Two other hexapeptides successfully found cancer-related human target proteins in reverse docking provided by PharmMapper [18]. HPEPDOCK was used to perform molecular docking for these hexapeptides [19]. The successful docking results provide evidence that the detected ACPs have a high possibility of exerting anti-cancer functions. Our study also determined the active site of the anticancer peptide through quantitative calculations. The approach we have used in this study integrated several bioinformatics software to provide ideas for discovering anticancer peptides from traditional Chinese medicine and exploring possible mechanisms. The resulting anticancer peptides, from foods that are homologous to medicine, are expected to play a practical role in the treatment of cancer. 

## 2. Results

### 2.1. ACP Screening

#### 2.1.1. Prediction and Screening of Anticancer Peptides of Sanghuang Proteins

The genomic and proteomic data of Sanghuang were obtained from NCBI (https://www.ncbi.nlm.nih.gov/protein, accessed on 20 August 2022). Sanghuang has a total gene length of 34.52 Mb and contains 62 proteins whose sequence information has been recorded. A chart of their length distribution is shown in Figure 2. Most of the proteins are between 200 and 1000 amino acid residues in length. The FASTA format of these peptides was entered into the input box at the center of mACPpred. None of them were identified as ACP.

Most of the discovered anticancer peptides are obtained by protein digestion. We used AntiCP 2.0 to scan proteins. Scanning results show that some of these proteins contains more than 10 ACP sequences inside. These sequences might be able to exert anticancer effects if accessible through digestion. 

#### 2.1.2. In Silico Digestion and Feature Prediction of Cleaved Fragments

Trypsin is the most commonly used enzyme, and the parental protein was cleaved with Expasy. We predicted the anticancer potential of the fragments using mACPpred. Nine of these peptides have anticancer probabilities that are all more than 0.60, making them all potential ACPs. The anticancer probabilities of nine peptides are shown in Table 1. In addition, AntiCP 2.0 was used to predict the properties of these peptides [12]. The predicted probabilities and properties of the anticancer peptides are also listed in Table 1. The vast majority of projected ACPs have either positive or zero charges. This demonstrates that they engage with negatively charged cancer cells to kill them through membrane contact, which is compatible with how some anticancer peptides interact with cancer cells [7]. This raises the credibility of the anticipated anticancer peptides. In addition, a study found that a negatively charged peptide can also act as a stabilizing and accumulating drug in the anticancer process [20]. We expect that the negatively charged anticancer peptides screened this time may also act in this way. 

### 2.2. Docking Study

#### 2.2.1. ACP Structure Prediction and Receptor Structure Obtained Based on Reverse Docking 

In our study, molecular docking was used to reveal the anticancer mechanism of ACP to corroborate the accuracy of anticancer peptide prediction. Therefore, it was necessary to find the binding mode of ACP to the target protein. The possible anticancer peptides in Sanghuang obtained by screening were blasted to find similar known anticancer peptide sequences. Similar known sequences can both confirm the anticancer potential of ACP and give relevant details about its mode of action. The CancerPPD database was used for sequence alignment [14]. Four peptides with high sequence identity to known anticancer peptides were screened from the nine peptides predicted to have anticancer potential from the website of the support vector machine. The results were shown in Table 2. 

The blast result of HALLAFTLGVR (protein No. 9) is the lipid bilayer of the cell membrane. Therefore, we did not perform molecular docking studies on protein No. 9. Blast results show that CHHNLTAAC (PTPRJ-pep19.7) is very similar to YIGAGLACSGLIGAGAGIGLIFSSLIASTAR (protein No. 2) in sequence. PTPRJ is a receptor protein tyrosinase, and PTPRJ-pep19.7 is capable of binding and activating PTPRJ to bring about anti-cancer effects [21]. Therefore, we expected that protein No. 2 might combine with PTPRJ to bring about an anti-cancer effect. The structure of PTPRJ was downloaded from the PDB website (PDB code: 2CFV [22]). It was adopted as the initial structure for molecular docking. No studies have indicated possible target proteins and mechanisms of action of the other two hexapeptides. Consequently, we consider protein No. 1 and protein No. 7 to be the research objects for reverse docking.

For reverse docking, we made use of the PharmMapper website [18]. Table 3 displayed the top 10 targets according to the normalized fit score of protein No. 1. We chose the protein that came in eighth place among them, and it is a human-derived protein linked to cancer. The protein (PDB code: 1N8Z [23]) is the extracellular domain of human HER2 complexed with Herceptin Fab. This complex had been linked to breast cancer [23]. Interestingly, as indicated in Table 4, this protein was also present in the top 10 targets of protein No. 7. So, for each of them, we used the extracellular domain of human HER2 complexed with Herceptin Fab [23] as the target protein.

The sequence with the highest score and lowest E value was protein No. 2. To determine the likely mechanism, the structure of this peptide was predicted using AlphaFold 2 [16]. Figure 3A is the resultant 3D structure. Its secondary structure was α helix. Because the sequences were too short, neither AlphaFold 2 nor SWISS-MODEL can forecast the structure of protein No. 1 and protein No. 7. Through NCBI, we discovered the ACP-containing sequence, and after structural simulation by AlphaFold, we used PyMOL to truncate the ACP part. The resulting structures of protein No. 1 and protein No. 7 are shown in Figure 3B,C, respectively. The secondary structures of both hexapeptides were random coils. 

#### 2.2.2. Molecular Docking of Receptors and Target Proteins and Quantum Chemical Calculation

For ACPs of different lengths, HDOCK and HPEPDOCK were selected for protein-peptide docking [17,19]. We used HDOCK for protein No. 2 [17]. The online site provides the top ten models for docking scores. We downloaded the docking model with the highest docking score and displayed it with visualization software, as shown in Figure 4. Figure 4A shows the location of ACP docking in the target protein. Figure 4B,C show the hydrogen bonding interaction between ACP and target protein. The carbon atoms of the target protein’s interacting residues are highlighted in pink. Hydrogen bonds are depicted in the illustration as yellow dotted lines. Protein No. 2’s G-19, S-23, and T-29 form four hydrogen bonds with the proteins T-8, D-11, and Y-1058 of PTPRJ. 

Molecular docking was carried out for both protein No. 1 and protein No. 7 using HPEPDOCK [19] in Figure 5 and Figure 6, respectively. Protein No. 1’s M-1and K-6 made four hydrogen bonds with the target protein’s Y-55, V-58, W-592, C-604, and I-606 (Figure 5B). Five hydrogen bonds were obtained between protein No. 7’s L-2, D-4, A-5 and the target protein’s R-332, Y-389, and Q-424 (Figure 6B). 

We obtained the HOMO and LUMO orbital maps of the two hexapeptides using the Gaussian 09 program. The orbital location reveals the distribution of active sites. Orbits indicated that the active site of protein No. 1 was located at M-1 and K-6 residues, shown in Figure 7A,B, consistent with the molecular docking results. The active site of protein No. 7 was located at D-4 and A-5, as shown in Figure 7C,D. The active site of protein 1 carried a positive charge, and the active site of protein 7 carried a negative charge. These matched the charge that was anticipated. Positively charged regions can help ACP bind to the surface of cancer cells, while negatively charged regions have the potential to play a role in stabilization or accumulation. The active sites of both hexapeptides were their concentrated sites for hydrogen bonding with the target protein.

## 3. Discussion

Cancer is a major public health problem worldwide. Since 1991, cancer mortality has progressively decreased due to breakthroughs in early identification, surgical methods, and tailored therapy [42]. Anticancer peptides, which are biologically active and have anticancer properties, have been discovered in various organisms. Food-derived ACPs, in particular, have considerable application potential due to their relative availability and inexpensive pricing [43].

Sanghuang, a traditional Chinese medicine, has proven anticancer effects. Relevant bioactive ingredients that have been studied include mannogalactan, polysaccharides, triterpenoids, pigments, flavonoids, and fatty acids [44]. The significance of peptides and proteins in Sanghuang has been disregarded for a very long period. Sanghuang has a long history of being ingested as food directly into patients to cure a wide range of illnesses. This shows that protein or peptides produced through enzymatic digestion may be used to exhibit their efficacy.

In order to efficiently and economically search for anticancer components from the Sanghuang proteome, we chose the virtual screening approach. Computer-aided drug design can provide significant savings in experimental costs during the initial stages of drug development and can reduce the chance of failure in the final stages [45]. High-quality databases and tools for the systematic evaluation of potential drug candidates both make computer-aided drug design more reliable and efficient [46]. The screening steps in this study sequentially included dataset processing, anticancer likelihood prediction, homology modeling, target protein prediction and molecular docking. We screened Sanghuang proteins downloaded from NCBI to confirm whether ACP is present. None of these proteins was ACP. They were subjected to computerized trypsin cleavage as parental proteins and the nine peptides produced were predicted to have high anti-cancer potential. Due to the presence of negatively charged molecules such as phosphatidylserine on the surface, cancer cells are primarily negatively charged. The great majority (88.90%) of anticipated ACPs have either positive or zero charges. The positive charge of the peptide increased its permeability through outer membranes and bactericidal action [7]. The ability of the cationic peptides to effectively cross the inner membrane’s acidic phospholipid composition points to the significance of electrostatic interactions [47]. Besides, the mechanism of action of anticancer peptides carrying negative charges has also been explained by relevant studies. Negatively charged anticancer peptides improve drug stability and help drug aggregation [20]. 

By blasting each peptide separately, four peptides that were very close to the current ACP were discovered. Due to technical limitations, we could only perform the next step of molecular docking for anti-cancer peptides with protein interactions. We selected three peptides that interact with proteins for structural simulations using AlphaFold 2 [16]. We searched for cancer-associated ligands of ACP by reverse docking or from references and used them for molecular docking with the identified peptides. Both hexapeptides can act on the extracellular domain of human HER2 complexed with Herceptin Fab. This target protein is associated with breast cancer [23]. Reliable binding modes raise the possibility that they have anticancer functions. Finally, the active sites of the two hexapeptides were predicted using quantum chemical calculations. The predicted active sites were exactly the sites where hydrogen bonds were formed during molecular docking.

Our research is an extension of the discovery of active substances in Sanghuang, first focusing on anticancer peptides. The existence of ACP can be called a new explanation for the anticancer function of Sanghuang. Our study can only indicate a few ways in which Sanghuang can fight cancer. The actual pathway through which ACPs bring about their anticancer effects in vivo needs to be investigated. Our study still provides an expectation that the use of Sanghuang as a health food in the treatment of breast cancer with Herceptin antibodies may be able to optimize the efficacy to some extent. The anticancer peptides obtained by screening are all predicted by computer and have not been verified through practical experiments. We have submitted our findings to the collaborating groups for practical experimental validation. Ligand-based computer-aided drug design can be limited by existing knowledge [48]. Data mining of anticancer peptides relies on known information about anticancer active molecules. Due to the mechanical limitations of the anti-cancer likelihood prediction database itself, research can only select anticancer peptides with similar sequence length, charge, affinity, and tertiary structure conformation from those of the CancerPPD database. 

The total number of proteins that have been sequenced in Sanghuang is small, and there are only three ACPs suitable for molecular docking. However, the research still provides a basic idea for the discovery of functional peptides from foods. For example, a database of anti-coronavirus peptides has recently been successfully constructed [49]. The same idea can be taken to screen traditional Chinese medicines with therapeutic effects. Edible fungi or traditional Chinese medicines containing anticancer peptides discovered through this method are promising as raw materials for drug extraction.

## 4. Materials and Methods

### 4.1. Anticancer Possibility Prediction of Peptides

This study used an online website, the mACPpred (http://thegleelab.org/mACPpred/ACPExample.html, accessed on 21 August 2022), to predict the anticancer potential of proteins [13]. Proteins and digested fragments in Sanghuang were scored. This website integrates seven feature-encoding-based models into a single model, exploiting different aspects of sequence information for sensitive prediction. The FASTA format of the protein to be predicted will be entered into the input box and submitted to obtain the prediction result of the anti-cancer potential.

### 4.2. In Silico Hydrolysis of the Parent Protein

Expasy online program was used to trypsinize potential parental proteins (https://web.expasy.org/peptide_cutter/, accessed on 21 August 2022) [50]. Trypsin is a peptide chain endonuclease with selective hydrolysis of arginine and lysine peptide chains. The peptide cutter software of Expasy offers a wide range of enzymes, including trypsin. After entering the protein sequence, the relative positions of the cleavage sites of different enzymes on the input protein will be annotated. After choosing the target enzyme, the acquired protein fragments will be displayed more correctly.

### 4.3. Characteristic Prediction of Anticancer Peptide

AntiCP2.0 predicts the hydrophobicity and charge of anticancer peptides (https://webs.iiitd.edu.in/raghava/anticp2/index.html, accessed on 21 August 2022) [12]. AntiCP2.0 is designed to predict and design anticancer peptides with high precision. The main dataset contains 861 anticancer peptides that have been experimentally validated, as well as 861 non-anticancer or antimicrobial peptides that have been validated.

### 4.4. Anticancer Peptide Database

The anticancer peptide database used in this paper is the CancerPPD database, created by Atul Tyagi, Abhishek Tuknait, Priya Anand, and others (http://crdd.osdd.net/raghava/cancerppd/, accessed on 21 August 2022) [14]. The authors compiled 3,491 ACP and 121 anticancer protein entries for the web resource from already published studies and patents. The anticancer peptide’s origin, characteristics, anticancer action, N- and C-terminal modifications, structure, and other details are all presented. The blast function of the database can be used to query known ACPs that are similar to the input sequence. 

### 4.5. Anticancer Peptide Structure Prediction

AlphaFold 2 is an artificial intelligence program of DeepMind designed to tackle the problem of protein folding (https://colab.research.google.com/github/deepmind/alphafold/blob/main/notebooks/AlphaFold.ipynb, accessed on 22 August 2022) [16]. The structure of the 31-residue-long peptide in the paper was predicted using AlpahFold 2. It is capable of predicting the protein structure of peptides of 16 residues and above. Users can enter the protein sequence and execute the program sequentially to obtain the structure. The two hexapeptides could not be directly predicted. We searched for proteins containing these two sequences, predicted their structures, and intercepted the regions of the hexapeptides.

### 4.6. Reverse Docking

Reverse docking is a key tool of computer-aided drug design [15]. The PharmMapper online tool uses reversed pharmacophore matching to compare the query chemical to a library of internal pharmacophore models in order to identify possible drug targets (http://lilab.ecust.edu.cn/pharmmapper/, accessed on 30 August 2022). We use this website to get the PDB codes of the top 300 likely target proteins of the hexapeptides. 

### 4.7. Quantitative Calculation

Gaussian 09 software was used to perform the quantum chemistry computations at the 6–31 G* set [51]. We obtained the HOMO and LUMO orbitals of the hexapeptides. By doing single-point calculations on the optimized geometries using the conductor-like polarized continuum model at the B3LYP 631G* level of theory, the consequences of the solution were assessed. For the computations, the surrounding solvent was assumed to have a value of 4. At 298.15 K and 1 atm pressure, frequency calculations were carried out to derive free energy adjustments. 

## 5. Conclusions

This study used bioinformatics methods to discover ACP from Sanghuang proteome data and to verify the anti-cancer mechanism. The results showed that nine fragments obtained by trypsin digestion had high anticancer potential. The ligand binding mechanism of three peptides was validated. The method used in the study can be applied to the data mining of other functional peptides from foods.

## Figures and Tables

**Figure 1 ijms-23-13682-f001:**
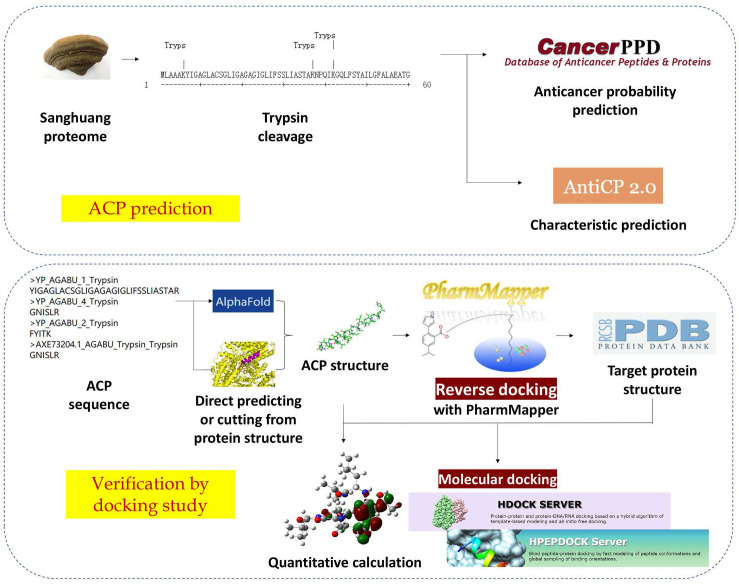
Schematic illustration of the steps involved in the in silico discovery of Sanghuang ACPs.

**Figure 2 ijms-23-13682-f002:**
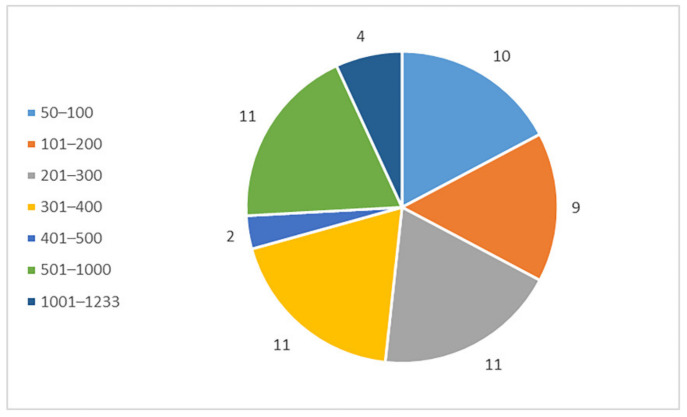
Distribution of the protein lengths of Sanghuang.

**Figure 3 ijms-23-13682-f003:**
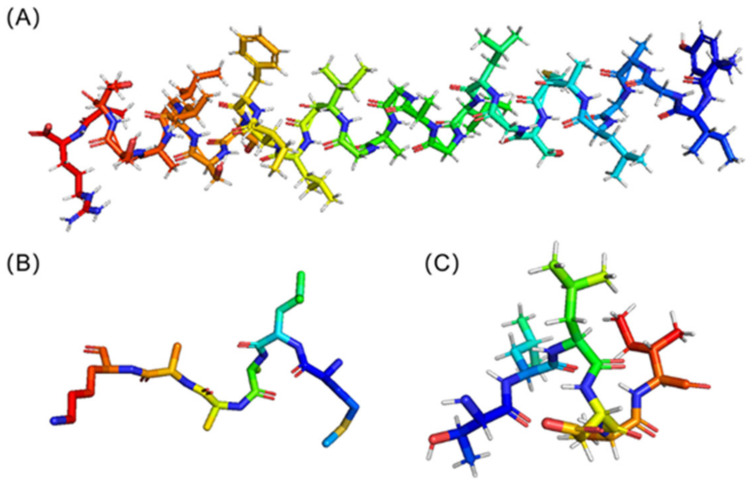
(**A**) The 3D Structure of Protein No. 2; (**B**) The 3D Structure of Protein No. 1; (**C**) The 3D Structure of Protein No. 7.

**Figure 4 ijms-23-13682-f004:**
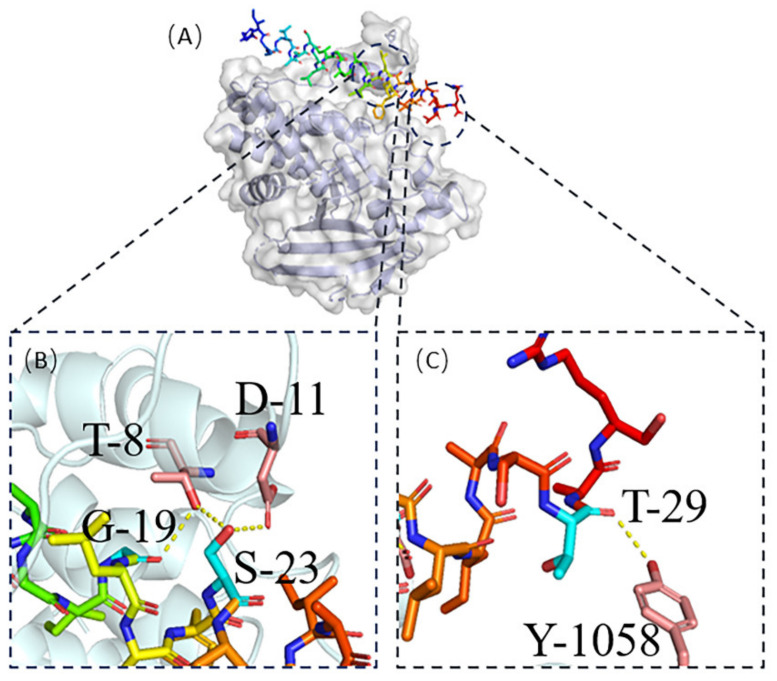
(**A**) Protein No. 2- PTPRJ complex; (**B**) The active residues around Protein No. 2 binding to PTPRJ (**C**) The active residues around Protein No. 2 binding to PTPRJ.

**Figure 5 ijms-23-13682-f005:**
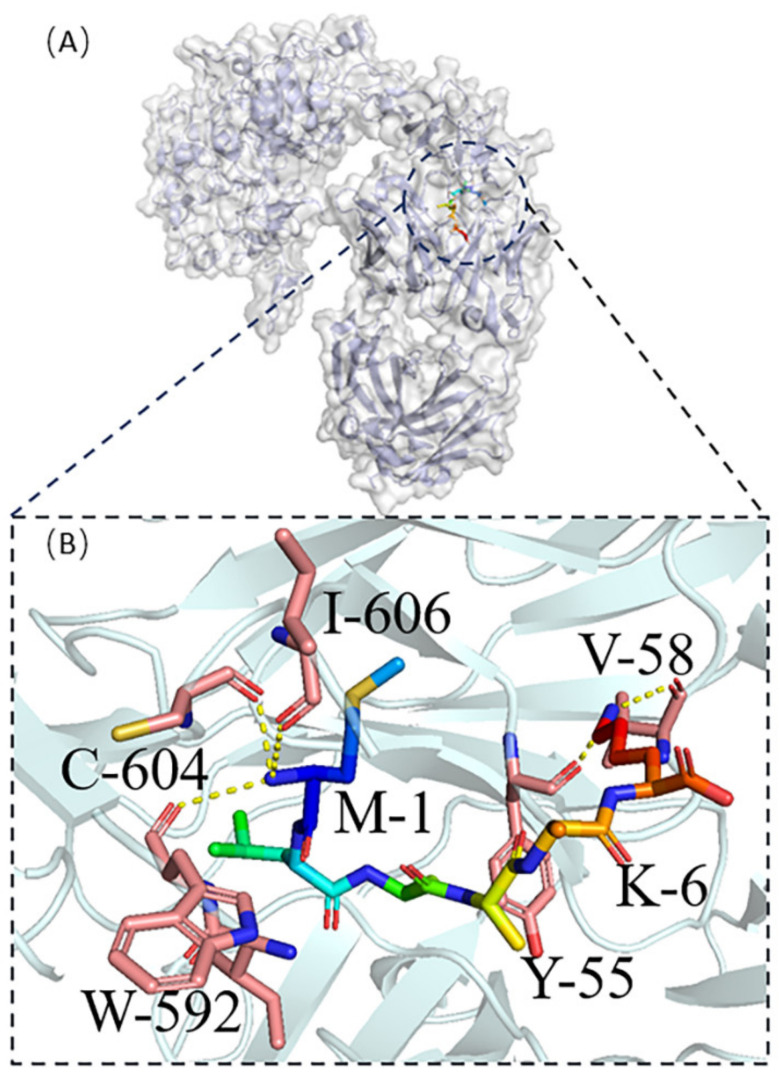
(**A**) Protein No. 1- HER2 complexes, (**B**) The active residues around Protein No. 1 binding to HER2 complexed with the Herceptin antigen-binding fragment.

**Figure 6 ijms-23-13682-f006:**
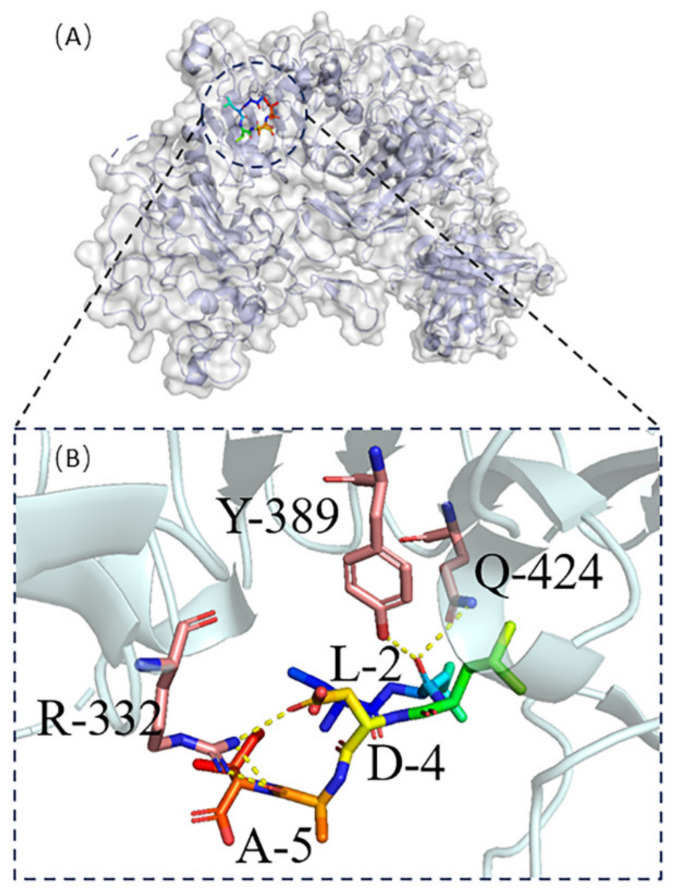
(**A**) Protein No. 7-HER2 complexes, (**B**) The active residues around Protein No. 7 binding to HER2 complexed with the Herceptin antigen-binding fragment.

**Figure 7 ijms-23-13682-f007:**
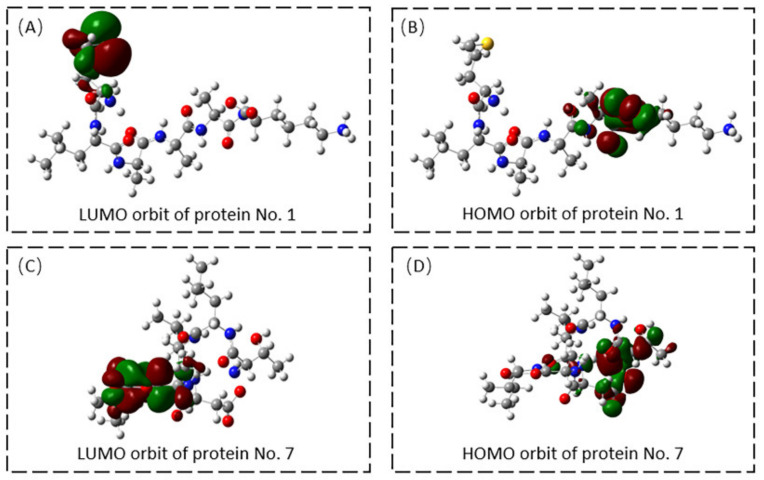
(**A**) LUMO orbit of Protein No. 1; (**B**) HOMO orbit of Protein No. 1; (**C**) LUMO orbit of Protein No. 7; (**D**) HOMO orbit of Protein No. 7.

**Table 1 ijms-23-13682-t001:** Properties of Selected Peptides for Synthesis.

No.	Sequence	Possibility	Charge	Hydrophobicity	Hydropathicity
1	MLAAAK	0.98	1.0	0.07	1.20
2	YIGAGLACSGLIGAGAGIGLIFSSLIASTAR	0.85	1.0	0.20	1.33
3	GNISLRS	0.80	1.0	−0.21	−0.24
4	MDTTK	0.83	0.0	−0.38	−1.38
5	VGYNPK	0.72	1.0	−0.18	−1.08
6	AGVVK	0.95	1.0	0.08	1.18
7	TLLDAI	0.61	−1.0	0.19	1.62
8	FYITK	0.97	1.0	0.02	0.28
9	HALLAFTLGVR	0.75	1.5	0.10	1.20

**Table 2 ijms-23-13682-t002:** Results of the blast of mACPpred to predict anticancer peptides in the CancerPPD database.

Sequence	Hit Sequence	Score	E Value
MLAAAK	GLLPCAESCVYIPCLTTVIGCSCKSKVCYKN	14	29
TLLDAI	MDSNKDERAYAQWVIIILHNVGSSPFKIANLGLSWGKLYADGNKDKEVYP	16	5
	SDYNGKTVGPDEKIQINSCGRENASSGTEGSFDIVDPNDGNKTIRHFYW		
	ECPWGSKRNTWTPSGSNTKWMVEWSGQNLDSGALGTITVDVLRKGN		
HALLAFTLGVR	KKWβ2,2WKK	13	110
YIGAGLACSGLIGAGAGIGLIFSSLIASTAR	CHHNLTAAC	18	3.3

**Table 3 ijms-23-13682-t003:** Top 10 targets of protein No. 1 and their function.

Rank	PDB ID	Target Name	Organism(s)	Function or Related Diseases	Fit Score	Normalized Fit Scores
1	4ESJ	NONE	*Streptococcus pneumoniae* TIGR4	DNA methylation-dependent restriction enzymes [24]	2.99	0.75
2	2CST	Aspartate aminotransferase, cytoplasmic	*Gallus*	related to liver function [25]	2.98	0.75
3	1NF0	Triosephosphate isomerase	*Saccharomyces cerevisiae*	glycolysis [26]	2.98	0.74
4	1B8H	DNA polymerase processivity component	*Escherichia phage* RB69	DNA replication [27]	2.98	0.74
5	2J04	Transcription factor tau 60 kDa subunit	*Saccharomyces cerevisiae*	transcription [28]	2.97	0.74
6	2NVB	NADP-dependent alcohol dehydrogenase	*Thermoanaerobacter brockii*	ethanol metabolism [29]	2.96	0.74
7	2ZJP	50S ribosomal protein L33	*Deinococcus radiodurans, Streptomyces actuosus*	translational regulation [30]	2.95	0.74
8	1N8Z	Receptor tyrosine-protein kinase erbB-2	*Mus musculus*, *Homo sapiens*	breast cancer [23]	2.95	0.74
9	2DHY	CUE domain-containing protein 1	*Homo sapiens*	ovarian cancer [31]	2.95	0.74
10	1JZ2	Ribonuclease HI	*Escherichia coli*	β-galactosidase reaction [32]	2.94	0.74

**Table 4 ijms-23-13682-t004:** Top 10 targets of protein No. 7 and their function.

Rank	PDB ID	Target Name	Organism(s)	Field of Application or Related Diseases	Fit Score	Normalized Fit Scores
1	3SQG	NONE	*Uncultured archaeon*	anaerobic oxidation of methane with sulphate [33]	7.07	0.79
2	1K99	Nucleolar transcription factor 1	*Homo sapiens*	transcription [34]	2.99	0.75
3	1KHV	Genome polyprotein	*Rabbit hemorrhagic disease virus*	rabbit hemorrhagic disease [35]	2.98	0.74
4	1N8Z	Receptor tyrosine-protein kinase erbB-2	*Mus musculus*, *Homo sapiens*	breast cancer [23]	2.98	0.74
5	1ZR4	Transposon gamma-delta resolvase	*Escherichia coli*	site-specific recombination [36]	2.96	0.74
6	2GH8	Capsid protein	*San Miguel sea lion virus* 4	a native calicivirus [37]	2.95	0.74
7	2YRV	AT-rich interactive domain-containing protein 4A	*Homo sapiens*	retinoblastoma [38]	2.95	0.74
8	1KCG	NKG2D ligand 3	*Homo sapiens*	trigger the NK cell lysis of various tumor and virally infected cells [39]	2.94	0.74
9	2VLD	UPF0286 protein PYRAB01260	*Pyrococcus abyssi*	endonuclease [40]	2.94	0.74
10	2QV3	Vacuolating cytotoxin	*Helicobacter pylori*	candidate antigens for Helicobacter pylori vaccines [41]	2.94	0.73

## Data Availability

Not applicable.

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
