# Peer review of "In Silico Discovery of Anticancer Peptides from Sanghuang"

_ijms, 2022, doi:10.3390/ijms232213682_

Round 1

Reviewer 1 Report

The article By Liu et al. illustrates the discovery of anticancer peptides from a fungus used in traditional Chinese medicine. I understand its focus is based on the "in silico" discovery of ACPs (this part that is well done and relatively complete). Still, I am not keen to accept a work based only on a theoretical base without any validation in vitro or ex-vivo. Therefore I would kindly ask the authors to provide data on apoptosis, antiproliferative test, MTT assays, analysis of cell cycle, and whatever else it may be done to prove their achieved results before accepting the paper.

Author Response

Point 1: The article By Liu et al. illustrates the discovery of anticancer peptides from a fungus used in traditional Chinese medicine. I understand its focus is based on the "in silico" discovery of ACPs (this part that is well done and relatively complete). Still, I am not keen to accept a work based only on a theoretical base without any validation in vitro or ex-vivo. Therefore I would kindly ask the authors to provide data on apoptosis, antiproliferative test, MTT assays, analysis of cell cycle, and whatever else it may be done to prove their achieved results before accepting the paper.

Response 1: We feel great thanks for your professional review work on our article. As you concerned, computers can only be used as an aid in the drug discovery process and must be experimentally proven when finally put into use. We have sent the experimental data to our partner group for wet experiments, and the next step of our work will be the in vitro and in vivo validation of the anticancer function. We politely suggest that the special issue to which we are submitting our paper is entitled Computer-Aided Screening and Action Mechanism of Bioactive Peptides. We would like to directly specifically draw attention to the crucial part that computation plays in anticancer peptide scanning. We sincerely expect that this is in accordance with the goals of this special edition. We already have excellent papers published in your journal that use exclusively theoretical computational approaches:

Xie, E., et al. (2022). "Using Computational Drug-Gene Analysis to Identify Novel Therapeutic Candidates for Retinal Neuroprotection." International Journal of Molecular Sciences 23(20): 12648. doi: 10.3390/ijms232012648

We appreciate your insightful remarks and thank you for your recognition of the theoretical calculation effort in this research.

Reviewer 2 Report

1. It is suggested to experimentally validate the affinity of the selected peptides. 

2. Molecular dynamics and free energy calculation for the selected peptides complex need to be performed. 

3. A flowchart demonstrating the complete work summary need to be provided. 

4. The methodology part needs to be revised, it's hard for the reader to understand. 

5. The computational screening of the library was the starting point of this study and it should be discussed more in the discussion. Authors may refer to several related articles:

DOI: 10.2174/1381612822666151125000550

DOI: 10.1007/s12272-015-0640-5

 DOI: 10.1093/bib/bbp023

6. The discussion part needs to be thoroughly revised. 

Author Response

Point 1: It is suggested to experimentally validate the affinity of the selected peptides.

Response 1: We feel great thanks for your professional review work on our article. As far as you are concerned, computers can only be utilized as a tool to help with the drug development process and must first undergo experimental validation before being deployed. We have sent the experimental data to our partner group for wet experiments, and the next step of our work will be the in vitro and in vivo validation of the anticancer function. We politely suggest that the special issue to which we are submitting our paper is entitled Computer-Aided Screening and Action Mechanism of Bioactive Peptides. We would like to directly specifically draw attention to the crucial part that computation plays in anticancer peptide scanning. We sincerely expect that this is in accordance with the goals of this special edition. We already have excellent papers published in your journal that use exclusively theoretical computational approaches:

Xie, E., et al. (2022). "Using Computational Drug-Gene Analysis to Identify Novel Therapeutic Candidates for Retinal Neuroprotection." International Journal of Molecular Sciences 23(20): 12648. doi: 10.3390/ijms232012648

Point 2: Molecular dynamics and free energy calculation for the selected peptides complex need to be performed.

Response 2: We are thankful for the kindly suggestions. Both molecular dynamics simulations and free energy calculations are qualitative rather than quantitative experiments. They tend to occur in pairs, and are more applicable when the protein-ligand relationship is one-to-many. We have found some relevant articles on molecular dynamics and free energy calculations that we hope will corroborate our ideas.

1, Heo L, Arbour CF, Janson G, Feig M. Improved Sampling Strategies for Protein Model Refinement Based on Molecular Dynamics Simulation. J Chem Theory Comput. 2021 Mar 9;17(3):1931-1943. doi: 10.1021/acs.jctc.0c01238

2, Matubayasi N. Free-energy analysis of protein solvation with all-atom molecular dynamics simulation combined with a theory of solutions. Curr Opin Struct Biol. 2017 Apr;43:45-54. doi: 10.1016/j.sbi.2016.10.005

However, our work intents to identify the anticancer peptides present in Sanghuang by analyzing the magnitude of their anticancer potential and other quantitative properties. Since the target proteins of these anticancer peptides were unknown before the study was performed, we could not find corresponding blank control molecules for qualitative comparison. Indeed, the aim of our study is simply to provide a way to screen for anticancer peptides from the large number of proteins contained in this medicinal fungus. Although we cannot perform free energy calculations, the docking scores we provide are indicative. We believe this is a sound and compelling research process for this type of data mining work. We have found some excellent literature on similar work. 

1, Zheng S, Zhu N, Shi C, Zheng H. Genomic data mining approaches for the discovery of anticancer peptides from Ganoderma sinense. Phytochemistry. 2020 Nov;179:112466. doi: 10.1016/j.phytochem.2020.112466

2, Kuzniak-Glanowska E, Glanowski M, Kurczab R, Bojarski AJ, Podgajny R. Mining anion-aromatic interactions in the Protein Data Bank. Chem Sci. 2022 Mar 1;13(14):3984-3998. doi: 10.1039/d2sc00763k

3, Han X, Shih J, Lin Y, Chai Q, Cramer SM. Development of QSAR models for in silico screening of antibody solubility. MAbs. 2022 Jan-Dec;14(1):2062807. doi: 10.1080/19420862.2022.2062807

With a better understanding of these target proteins, we are more than willing to continue our research on the anticancer peptides mentioned in this paper in future work.  We sincerely thank you for your professional advice.

Point 3: A flowchart demonstrating the complete work summary need to be provided.

Response 3: Thank you very much for your comments. Our flowchart has included all the steps of the research effort, but we regret that it does not adequately convey the intended message. The cue words in the graphic have been rearranged. We increased the font size and bolded the text that was not clear enough to read. Key steps have also been highlighted. Please see the revised version.

Point 4: The methodology part needs to be revised, it's hard for the reader to understand.

Response 4: We apologize for not being clear about the methodology and materials. We have carefully revised the sequencing and logic issues. Please see the revised version.

Line 269 and line 285: We changed the subheading to “Anticancer possibility prediction of peptides” and “Characteristic prediction of anticancer peptide”.

Line 270 to line 276: We have revised this whole paragraph as “This study used an online website, the mACPpred (http://thegleelab.org/mACPpred/ACPExample.html), to predict the anticancer potential of proteins [13]. Proteins smaller than 100 residues and digested fragments in Sanghuang were scored. This website integrates seven feature-encoding-based models into a single model, exploiting different aspects of sequence information for sensitive prediction. The FASTA format of the protein to be predicted will be entered into the input box and submitted to obtain the prediction result of the anti-cancer potential.”

Line 279 to 283:  We have revised the middle part of this paragraph as “Trypsin is a peptide chain endonuclease with selective hydrolysis of arginine and lysine peptide chains. The peptide cutter software of Expasy offers a wide range of enzymes, including trypsin. After entering the protein sequence, the relative positions of the cleavage sites of different enzymes on the input protein will be annotated.”

Line 292: For fluency, we have revised the first sentence to “The anticancer peptide database used in this paper is the CancerPPD database”

Line 305 to 306: We have changed the sentence with an ambiguous expression to “It is capable of predicting the protein structure of peptides of 16 residues and above.”

Line 309: We have replaced “truncated” with “intercepted”

Point 5: The computational screening of the library was the starting point of this study and it should be discussed more in the discussion. Authors may refer to several related articles:

DOI: 10.2174/1381612822666151125000550

DOI: 10.1007/s12272-015-0640-5

DOI: 10.1093/bib/bbp023

Response 5: After careful reading, we found that the references provided by the reviewers were very helpful and enriched our discussion. As suggested by the reviewer, we have added more references in the discussion part.

Line 215 to 222: We have added the arguments as “In order to efficiently and economically search for anticancer components from the Sanghuang proteome, we chose the virtual screening approach. Computer-aided drug design can provide significant savings in experimental costs during the initial stages of drug development and can reduce the chance of failure in the final stages [47]. High-quality databases and tools for the systematic evaluation of potential drug candidates both make computer-aided drug design more reliable and efficient [48]. The screening steps in this study sequentially included data set processing, anticancer likelihood prediction, homology modeling, target protein prediction and molecular docking.”

Line 254 to 260: We have added the arguments as “We have submitted our findings to the collaborating groups for practical experimental validation. Ligand-based computer-aided drug design can be limited by existing knowledge [50]. Data mining of anticancer peptides relies on known information about anticancer active molecules. Due to the mechanical limitations of the anti-cancer likelihood prediction database itself, the research can only select anticancer peptides with similar sequence length, charge, affinity, and tertiary structure conformation from those of the CancerPPD database.”

Point 6: The discussion part needs to be thoroughly revised.

Response 6: We appreciate your comments on the discussion section, and additional significant adjustments have been implemented. We hope we have more clearly discussed the subject.

Line 202: We have changed “the risk of death” to “cancer mortality”

Line 222 to 227: We have reorganized the language in this section as “We screened Sanghuang proteins downloaded from NCBI to confirm whether ACP is present. None of these proteins was ACP. They were subjected to computerized trypsin cleavage as parental proteins and 9 produced peptides were predicted to have high anti-cancer potential. Due to the presence of negatively charged molecules like phosphatidylserine on the surface, cancer cells are primarily negatively charged.”

Line 231 to 234: We have added the following arguments “Besides, the mechanism of action of anticancer peptides carrying negative charges has also been explained by relevant studies. Negatively charged anticancer peptides improve drug stability and help drug aggregation [21].”

Line 236 to 238: We have reorganized the language in this section as “Due to technical limitations, we could only perform the next step of molecular docking for anti-cancer peptides with protein interactions. We selected three peptides that interact with proteins for structural simulations using AlphaFold 2 [43]”

Line 248 to 252: We have added the following arguments “Our study can only indicate a few ways in which Sanghuang can fight cancer. The actual pathway through which ACPs exert their anticancer effects in vivo needs to be deciphered. Our study still provides an expectation that the use of Sanghuang as a health food in the treatment of breast cancer with Herceptin antibodies may be able to optimize the efficacy to some extent.”

Round 2

Reviewer 1 Report

The authors still do not validate their findings by in vitro experiments. This can be accepted if we consider a merely computational article but according to my opinion an high impact journal such as IJMS must publish complete works, that includes every single step of a discovery, from the project to the validation of the findings. I recommend to find a different journal where to publish this article.

Reviewer 2 Report

The authors have mostly addressed the raised comments.